# Deep Stochastic Processes
# via Functional Markov Transition Operators

**Jin Xu**[1][*]    **Emilien Dupont**[1][†]    **Kaspar Märtens**[2]    **Tom Rainforth**[1]    **Yee Whye Teh**[1][‡]

[1] Department of Statistics, University of Oxford, UK.
[2] Big Data Institute, University of Oxford, UK.

## Abstract

We introduce Markov Neural Processes (MNPs), a new class of Stochastic Processes (SPs) which are constructed by stacking sequences of neural parameterised Markov transition operators in function space. We prove that these Markov transition operators can preserve the exchangeability and consistency of SPs. Therefore, the proposed iterative construction adds substantial flexibility and expressivity to the original framework of Neural Processes (NPs) without compromising consistency or adding restrictions. Our experiments demonstrate clear advantages of MNPs over baseline models on a variety of tasks.

## 1 Introduction

Stochastic Processes (SPs) are widely used in many scientific disciplines, including biology [3], chemistry [57], neuroscience [34], physics [45] and control theory [1]. They are formed by a (typically infinite) collection of random variables and can be used to model data by considering the conditional distribution of target variables given observed context variables. In machine learning, SPs in the form of Bayesian nonparametric models—such as Gaussian Processes (GPs) [47] and Dirichlet processes [56]—are used in tasks such as regression, classification, and clustering. SPs parameterised by neural networks have also been used for meta-learning [21, 60] and generative modelling [15, 39].

With the increasing amount of data available, and the complex patterns arising in many applications, more flexible and scalable SP models with greater learning capacity are required. The Neural Process (NP) family [18, 21, 22, 25, 30] meets this demand by parameterising SPs with neural networks, and enjoys greater flexibility and computational efficiency compared to traditional nonparametric models.

Unfortunately, the original version of NPs [22] lacks expressivity and often underfits the data in practice [30]. Various extensions have therefore been proposed—such as Attentive Neural Processes (ANPs) [30], Convolutional Neural Processes (CONVNPs) [18, 25] and Gaussian Neural Processes (GNPs) [4]—to improve expressivity. However, these models—which we refer to as *predictive* SPs—are based around directly constructing mappings from contexts to predictive distributions, and therefore forgo the fully generative nature of the original NP formulation. This can be problematic as it means that they are no longer *consistent* under conditioning: their predictive distributions no longer correspond to the conditional distribution of an underlying SP prior. In turn, this can cause a variety of issues, such as conflicts or inconsistencies between predictions and miscalibrated uncertainties.

To address these issues, we propose an alternative mechanism to extend the NP family and provide increased expressivity while maintaining their original fully generative nature and consistency. We

---

[*]Corresponding author: `<jin.xu@stats.ox.ac.uk>`
[†]ED is now at Google DeepMind; this work was done while ED was at Oxford.
[‡]YWT is at both Google DeepMind and Oxford; this work was done at Oxford.

37th Conference on Neural Information Processing Systems (NeurIPS 2023).

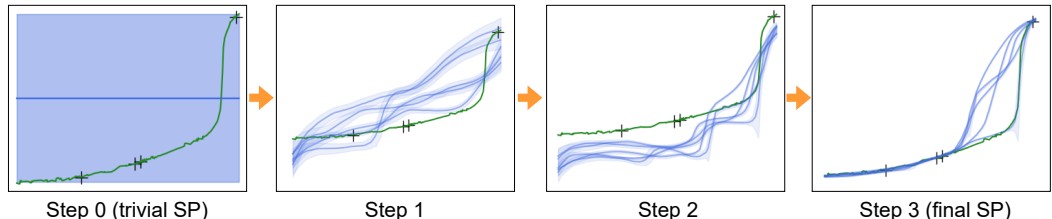

| Step 0 (trivial SP) | Step 1 | Step 2 | Step 3 (final SP) |

Figure 1: **MNPs construct expressive SPs via iterative transformations.** Here we use MNPs to model distributions over monotonic functions. We start with trivial initial SPs and gradually transform them into more complex SPs conditioned on observed context points marked as black crosses. The blue lines represent sampled mean functions and the shaded region indicates one standard deviation.

begin by generalising the marginal density functions of NPs. The generalised form can be viewed as Markov transition operators with functions as states. This lays the foundation for stacking these operators to construct more powerful generative SP models that we call Markov Neural Processes (MNPs). MNPs can be seen as Markov chains in *function* space, parameterised by neural networks: they make iterative transformations from simple initial SPs into more flexible, yet properly defined, SPs (as illustrated in Figure 1), without compromising consistency or introducing additional assumptions.

To empirically demonstrate the value of our proposed approach, we first benchmark MNPs against baselines on 1D function regression tasks, and conduct ablation studies in this controlled setting. We then show that they can be used as a high-performing surrogate for contextual bandit problems. Finally, we apply them to geological data, for which they demonstrate encouraging performance.

## 2  Background

A SP is a (typically infinite) collection of random variables defined on a common probability space. We can consider a SP as a random function $F : \mathcal{X} \to \mathcal{Y}$ where inputs can be regarded as indexing the output random variables. With a relaxed use of notation, we employ $p(f)$ in denoting a SP, where $f$ maps inputs $x \in \mathcal{X}$ to outputs $y \in \mathcal{Y}$. Kolmogorov's Consistency Theorem shows that a SP can be *indirectly* defined via a collection of marginal distributions, $\{p_{x_{1:n}}(y_{1:n})\}_{x_{1:n} \in \mathcal{X}^n}$ (we drop $\mathcal{X}^n$ from here on for conciseness) if they, for any permutation $\pi$ and all possible sets of inputs $x_{1:n} \in \mathcal{X}^n$, satisfy the *exchangeability* condition:

$$p_{x_{1:n}}(y_{1:n}) = p_{\pi(x_{1:n})}(\pi(y_{1:n})) := p_{x_{\pi(1)},\ldots,x_{\pi(n)}}(y_{\pi(1)},\ldots,y_{\pi(n)}), \tag{1}$$

and the *(marginal) consistency* condition:

$$p_{x_{1:m}}(y_{1:m}) = \int p_{x_{1:n}}(y_{1:n})\, \mathrm{d}y_{m+1:n} \quad \forall 1 \le m < n. \tag{2}$$

If none of the random variables in a SP are observed, we call it a *prior* SP. For any two distinct subsets of datapoints $\mathcal{C} = \{(x_i, y_i)\}_{i=1}^m$ (the context) and $\mathcal{T} = \{(x_i, y_i)\}_{i=m+1}^n$ (the target), one can use this prior SP to compute the conditional density $p_{x_{m+1:n}|x_{1:m}}(y_{m+1:n}|y_{1:m})$ via Bayesian inference. If inference is exact, it can be proved that the collection of conditional distributions $\{p_{x_{m+1:n}|x_{1:m}}(y_{m+1:n}|y_{1:m})\}_{x_{m+1:n}}$ also satisfy exchangeability and consistency, and hence define a valid *posterior* SP, $p(f|\mathcal{C})$. We discuss the impact of approximate inference in Section 4.3.

Based on a conditional version of de Finetti's Theorem, a NP [22] defines a SP by providing a collection of exchangeable and consistent marginal densities parameterized by neural networks. Specifically, they set up their marginal densities to have the form:

$$p_{x_{1:n}}(y_{1:n}; \theta) = \int p_\theta(z) \prod_{i=1}^n \mathcal{N}(y_i|\mu_\theta(x_i, z), \sigma_\theta^2(x_i, z))\mathrm{d}z, \tag{3}$$

where $z$ is a latent variable which captures dependencies across different input locations, $y_i := f(x_i)$, $\mu_\theta$ and $\sigma_\theta$ are deep neural networks, and $p_\theta(z)$ is a, typically Gaussian, prior distribution on the latents. Note that this form is more general than the one in [22] where both $p_\theta$ and $\sigma_\theta$ are not learnt.

## 3 Generative versus predictive stochastic process models

Before introducing our MNP approach, we first delve into the distinctions between conventional, consistent, SP models like NPs, and the predictive SP models corresponding to popular NP extensions, such as ANPs, CONVNPs, and GNPs, highlighting some of the potential drawbacks with the latter.

We introduce the term *generative* SP model to refer to the standard case where one first specifies a prior SP, $p(f)$, and then relies on Bayesian inference to compute posterior a SP, $p(f|\mathcal{C})$. The context and the target are only distinguished for inference, and they are treated equally in model specification. Traditional nonparametric models such as GPs, Student-t processes and the original NPs belong to this category. Since posterior SPs are inferred under the same prior SP, under the condition of exact inference, for two different contexts $\mathcal{C} = \{(x_i, y_i)\}_{i \in \mathcal{C}}$ and $\mathcal{C}' = \{(x_i, y_i)\}_{i \in \mathcal{C}'}$, we have

$$\int p(f|\mathcal{C})p_{x_\mathcal{C}}(y_\mathcal{C})\mathrm{d}y_\mathcal{C} = \int p(f|\mathcal{C}')p_{x_{\mathcal{C}'}}(y_{\mathcal{C}'})\mathrm{d}y_{\mathcal{C}'}, \tag{4}$$

where $x_\mathcal{C} := \{x_i\}_{i \in \mathcal{C}}$ and $x_{\mathcal{C}'} := \{x_i\}_{i \in \mathcal{C}'}$ are taken as fixed, and $p_{x_\mathcal{C}}(y_\mathcal{C})$ and $p_{x_{\mathcal{C}'}}(y_{\mathcal{C}'})$ are defined through the prior SP $p(f)$. We call this property *conditional consistency* to set apart from marginal consistency defined in Equation (2). Note that conditional consistency is a prerequisite for marginal consistency to hold for the prior $p(f)$: if the two integrals in Equation (4) are not equal, at least one must be different to the prior distribution $p(f)$.

By contrast, predictive SP models directly construct mappings from a context $\mathcal{C}$ to a predictive SP $p(f; \mathcal{C})$ and make a clear distinction between the context and the target in model specification. Consequently, for these models $p(f; \mathcal{C})$ no longer corresponds to a posterior derived from a certain prior $p(f)$, so they no longer need to satisfy conditional consistency. As such, they *no longer form a valid conditioning of a* SP: though they are typically constructed to ensure $p(f; \mathcal{C})$ is itself a valid SP for new evaluations of $f$ with $\mathcal{C}$ held fixed, this SP is no longer itself derived as the conditional of a prior SP. As such, predictive SP models are not consistent under updating and no longer update their uncertainties in a Bayesian manner as new data is observed. In short, they are *no longer treating the data itself as being drawn from an* SP.

Though predictive SP models have proven to be effective tools for meta-learning tasks, such as 1-D regression, image regression, and few-shot image classification [4, 18, 30, 60], this lack of consistent updating can be problematic in scenarios where the context is not fixed, such as when performing sequential updating. In particular, their uncertainties will be mismatched with how the model is updated in practice as new data is observed. This has the potential to be problematic for a wide variety of possible applications involving sequential decision-making, such as Bayesian experimental design [7, 46], active learning [2, 27, 51], Bayesian optimization [19, 23], and contextual bandit problems [35, 50]. There is also the potential for such models to fall foul of so-called Dutch Book scenarios [26], wherein the inconsistencies of the model can lead to conflicting predictions.

## 4 Markov Neural Processes

To correct the shortfalls of NPs while maintaining their conditional consistency, we now introduce a more expressive family of generative SP models termed Markov Neural Processes (MNPs). Our starting point is to extend NP density functions into a generalised form that can be viewed as a transition operator which transforms a trivial SP to a more flexible one. MNPs are then formed by stacking sequences of these transition operators to form a highly expressive and flexible model class. The training and inference procedure for MNP mirrors that of NP, but we also introduce a novel inference model that allows for efficient learning in our scenario.

### 4.1 A more general form of Neural Process density functions

Recall the form of the NP marginal density functions from Equation (3). One can draw joint samples from $p_{x_{1:n}}(y_{1:n}; \theta)$ via reparameterization using:

$$y_{1:n}^{(0)} \sim \mathcal{N}(\mathbf{0}, \mathbf{1}), \quad z \sim p_\theta(z), \quad y_i = \sigma_\theta(z, x_i) \cdot y_i^{(0)} + \mu_\theta(z, x_i). \tag{5}$$

The key starting point for MNPs is to show that this can be generalised to the case where $y_{1:n}^{(0)}$ is drawn from any given SP of its own, $p(f^{(0)})$, and each $y_i$ is any invertible transformation, $F_\theta$, of

$y_i^{(0)}$, parameterized by $x_i$ and $z$. Specifically, we introduce the following result (see Appendix A for proof).

**Proposition 4.1.** *Let $F_\theta(\cdot; x, z) : \mathcal{Y} \mapsto \mathcal{Y}$ denote an invertible transformation between outputs, parameterized by the input and latent. If $\{p_{x_{1:n}}(y_{1:n}^{(0)})\}_{x_{1:n}}$ is a valid SP (i.e. it satisfies Equation (1) and Equation (2)) and*

$$y_{1:n}^{(0)} \sim \{p_{x_{1:n}}(y_{1:n}^{(0)})\}_{x_{1:n}}, \quad z \sim p_\theta(z), \quad y_i = F_\theta(y_i^{(0)}; x_i, z). \tag{6}$$

*then $\{p_{x_{1:n}}(y_{1:n})\}_{x_{1:n}}$ is also a valid SP.*

Our next step is to realise that Equation (6) can be viewed as a Markov transition in function space, denoted as $p(f|f^{(0)})$, which transforms a simpler SP $p(f^{(0)})$ to a more expressive one $p(f)$. We can thus generalise things further by stacking sequences of Markov transition operators in function spaces (FMTOs), denoted by $p(f^{(1)}|f^{(0)}), \ldots, p(f^{(T)}|f^{(T-1)})$, together to form a Markov chain $f^{(0)} \to \cdots \to f^{(T)}$ in function space. This will be the basis of MNPs.

## 4.2 Markov chains in function space

Analogously to defining a SPs through its marginals, we indirectly specify FMTOs using a collection of marginal Markov transition operators (MMTOs), denoted by $\{p_{x_{1:n}}(y_{1:n}|y_{1:n}^{(0)})\}_{x_{1:n}}$, where each MMTO is just Markov transition operator over a finite sequence of function outputs.

To adapt the definitions of consistency and exchangeability for SPs to the transition operator setting, we say that the MMTOs are consistent if and only if, for any $1 < m < n$ and sequence $x_{1:n} \in \mathcal{X}^n$,

$$\int p_{x_{1:n}}(y_{1:n}|y_{1:n}^{(0)}) \, \mathrm{d}y_{m+1:n} = p_{x_{1:m}}(y_{1:m}|y_{1:n}^{(0)}) = p_{x_{1:m}}(y_{1:m}|y_{1:m}^{(0)}). \tag{7}$$

Similarly, MMTOs are exchangeable if, and only if, for all possible permutations $\pi$,

$$p_{x_{1:n}}(y_{1:n}|y_{1:n}^{(0)}) = p_{\pi(x_{1:n})}(\pi(y_{1:n})|\pi(y_{1:n}^{(0)})). \tag{8}$$

Note that, if we consider $(x_i, y_i^{(0)})$ as inputs, and $y_i$ as function outputs, the transition operator $p_{x_{1:n}}(y_{1:n}|y_{1:n}^{(0)})$ can be seen as the finite marginals of a random function $F' : \mathcal{X} \times \mathcal{Y} \to \mathcal{Y}$ whose distribution is a SP, and the conditions (7) and (8) correspond to (2) and (1).

Provided that these MMTOs are consistent and exchangeable, the FMTOs in function space will also be well-defined indirectly—i.e. the transition produces a well-defined SP given input random functions from a SP—as per the following result (see Appendix A for proof):

**Proposition 4.2.** *If the collection of marginals before transition $\{p_{x_{1:n}}(y_{1:n}^{(0)})\}_{x_{1:n}}$ is consistent and exchangeable (as per Equations (1) and (2)) and the collection of MMTOs $\{p_{x_{1:n}}(y_{1:n}|y_{1:n}^{(0)})\}_{x_{1:n}}$ is also consistent and exchangeable (as per Equations (7) and (8)), then the collection of marginals after transition $\{p_{x_{1:n}}(y_{1:n})\}$ is also consistent and exchangeable, hence defining a valid SP.*

Furthermore, a Markov chain in function space can be constructed by a sequence of FMTOs $p(f^{(1)}|f^{(0)}), \ldots, p(f^{(T)}|f^{(T-1)})$ where $p(f^{(t)}|f^{(t-1)}) := \{p_{x_{1:n}}(y_{1:n}^{(t)}|y_{1:n}^{(t-1)})\}_{x_{1:n}}$. With repeated applications of Proposition 4.2, if the initial state $\{p_{x_{1:n}}(y_{1:n}^{(0)})\}_{x_{1:n}}$ is exchangeable and consistent, $\{p_{x_{1:n}}(y_{1:n}^{(T)})\}_{x_{1:n}}$ at time $T$ is also exchangeable and consistent, hence defining a SP $p(f^{(T)})$.

Equation (6) then provides a valid construction of consistent and exchangeable MMTOs by introducing an auxiliary latent variable $z$. The transition operator is written as

$$p_{x_{1:n}}(y_{1:n}|y_{1:n}^{(0)}; \theta) = \int p_\theta(z) \prod_{i=1}^{n} \delta(y_i - F_\theta(y_i^{(0)}; x_i, z)) \, \mathrm{d}z, \tag{9}$$

where both $p_\theta$ and $F_\theta$ in Equation (9) can be parameterised by neural networks. Critically, we can extend Proposition 4.2 to cover these auxiliary settings in Equation (9) as well, as per the following result (see Appendix A for proof):

**Proposition 4.3.** MMTOs *in the form of Equation (9) are consistent and exchangeable.*

## 4.3 Parameterisation, inference and training

We can now define a MNP as a sequence of neural FMTOs, with each FMTO indirectly specified via a collection of MMTOs in the form of Equation (9). If we specify a distribution over the sequence of auxiliary latent variables $p_\theta(z^{(1:T)})$ along with an initial SP with marginals $p_{x_{1:n}}(y_{1:n}^{(0)})$, we can write down the marginal distribution over function outputs $y_{1:n} := y_{1:n}^{(T)}$ for a sequence of inputs $x_{1:n}$ under the MNP model as (see Figure 2a for illustration and Appendix A for derivation):

$$p_{x_{1:n}}(y_{1:n}; \theta) = \int p_\theta(z^{(1:T)}) p_{x_{1:n}}(y_{1:n}^{(0)}) \prod_{t=1}^{T} \prod_{i=1}^{n} p_\theta^{(t)}(y_i^{(t)} \mid y_i^{(t-1)}, x_i, z^{(t)}) \mathrm{d}y_{1:n}^{(0:T-1)} \mathrm{d}z^{(1:T)} \quad (10)$$

$$= \int p_\theta(z^{(1:T)}) p_{x_{1:n}}(y_{1:n}^{(0)}) \prod_{t=1}^{T} \prod_{i=1}^{n} \left| \det \frac{\partial F_\theta^{(t)}(y_i^{(t-1)}; x_i, z^{(t)})}{\partial y_i^{(t-1)}} \right| \mathrm{d}z^{(1:T)}$$

According to Propositions 4.2 and 4.3, $\{p_{x_{1:n}}(y_{1:n}; \theta)\}_{x_{1:n}}$ defines a valid SP parameterised by $\theta$. The initial SP can be arbitrarily chosen, as long as we can evaluate its marginals $p_{x_{1:n}}(y_{1:n}^{(0)})$. In our experiments, we use a trivial SP where all the outputs are i.i.d. standard normal distributed. We adopt normalising flows [16, 43, 49] to parameterise the invertible transformations $F_\theta^{(t)}$.

To integrate over latent variables $z^{(1:T)}$ in Equation (10), we introduce a posterior approximation $q_{x_{1:n}}(z^{(1:T)} | y_{1:n}; \phi)$. For many applications, we need to learn and query the conditional distributions of a target $\mathcal{T} = \{(x_i, y_i)\}_{i=m+1}^{n}$ given context $\mathcal{C} = \{(x_i, y_i)\}_{i=1}^{m}$. To better reflect the desired model behaviour at test time, similar to NPs [22], we train the model by maximising the following approximation of the conditional log-likelihood w.r.t. both model $\theta$ and variational $\phi$ parameters:

$$\mathcal{L}_{\theta, \phi}(y_{1:n}; x_{1:n}) :=$$
$$\mathbb{E}_{q_{x_{1:n}}(z^{(1:T)} \mid y_{1:n}; \phi)} \left[ \log p_{x_{m+1:n}}(y_{m+1:n} \mid z^{(1:T)}, \theta) + \log \frac{q_{x_{1:m}}(z^{(1:T)} \mid y_{1:m}; \phi)}{q_{x_{1:n}}(z^{(1:T)} \mid y_{1:n}; \phi)} \right] \quad (11)$$

where $\log p_{x_{m+1:n}|x_{1:m}}(y_{m+1:n} \mid y_{1:m}; \theta) \gtrsim \mathcal{L}_{\theta, \phi}(y_{1:n}; x_{1:n})$. Here $q_{x_{1:m}}(z^{(1:T)} \mid y_{1:m}; \phi)$ can be seen as an approximate prior conditioned on the context $\{(x_i, y_i)\}_{i=1}^{m}$, while $q_{x_{1:n}}(z^{(1:T)} \mid y_{1:n}; \phi)$ is the approximate posterior after the target $\{(x_i, y_i)\}_{i=m+1}^{n}$ is observed. Thus the prior and the posterior share the same inference model with parameters $\phi$. The training objective is optimised by stochastic gradient descent, and the gradients $\nabla_{\theta, \phi} \mathcal{L}_{\theta, \phi}(y_{1:n}; x_{1:n})$ can be efficiently estimated with low variance using the reparameterisation trick [40].

However, different from NPs, the latent variables $z^{(1:T)}$ for MNPs are a sequence. Therefore, we propose a different inference model which shares parameters with the generative model above. Firstly, we write $q_{x_{1:m}}(z^{(1:T)} \mid y_{1:m}; \phi)$ in a factorised form:

$$q_{x_{1:n}}(z^{(1:T)} \mid y_{1:n}; \phi) = \prod_{t=1}^{T} q_{x_{1:n}}^{(t)}(z^{(t)} | z^{(t+1)}, y_{1:n}^{(t)}; \phi) \quad (12)$$

where $\phi$ are variational parameters and we set $z^{(T+1)} = \mathbf{0}$, $y_{1:n} = y_{1:n}^{(T)}$. In our implementation, we use a Gaussian distribution for each factor $q_{x_{1:n}}^{(t)}$, with mean $\mu_\phi^{(t)}(z^{(t+1)}, y_{1:n}^{(t)}, x_{1:n})$ and variance $\Sigma_\phi^{(t)}(z^{(t+1)}, y_{1:n}^{(t)}, x_{1:n})$), Here $\mu_\phi^{(t)}$ and $\Sigma_\phi^{(t)}$ are parameterised functions invariant to the permutation of data points $\{(x_i, y_i)\}_{i=1}^{n}$, i.e.

$$\mu_\phi^{(t)}(z^{(t+1)}, y_{1:n}^{(t)}, x_{1:n}) = \mu_\phi^{(t)}(z^{(t+1)}, y_{\pi(1:n)}^{(t)}, x_{\pi(1:n)})$$
$$\Sigma_\phi^{(t)}(z^{(t+1)}, y_{1:n}^{(t)}, x_{1:n}) = \Sigma_\phi^{(t)}(z^{(t+1)}, y_{\pi(1:n)}^{(t)}, x_{\pi(1:n)}).$$

We can parameterise them by first having

$$r^{(t)} = \text{SETENCODER}(y_{1:n}^{(t)}, x_{1:n})) \quad (13)$$

where SETENCODER could be a Set Transformer [36], Deep Sets [61], then taking

$$\mu_\phi^{(t)}(z^{(t+1)}, y_{1:n}^{(t)}, x_{1:n}) = \text{MLP}_\mu(z^{(t+1)}, r^{(t)}) \quad (14)$$

$$\sigma_\phi^{(t)}(z^{(t+1)}, y_{1:n}^{(t)}, x_{1:n}) = \text{MLP}_\sigma(z^{(t+1)}, r^{(t)}). \quad (15)$$

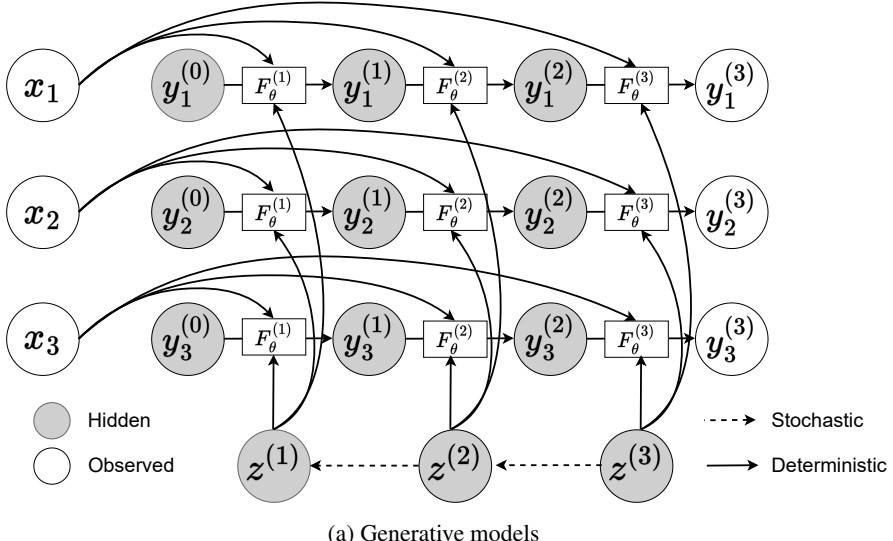

(a) Generative models

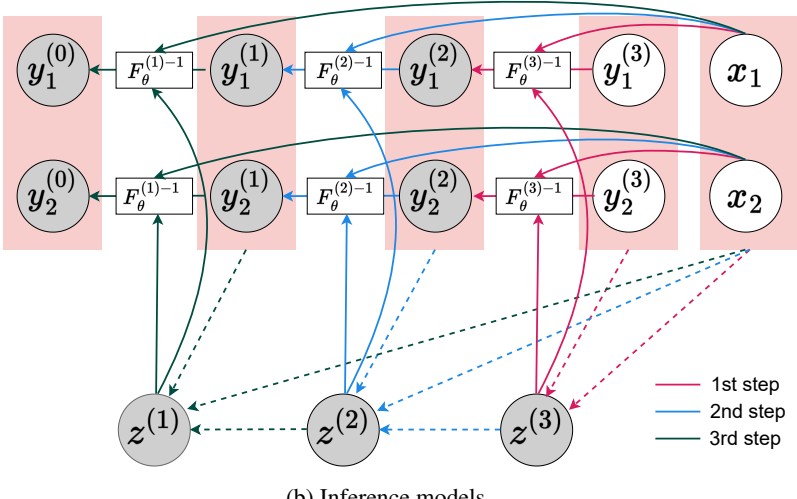

(b) Inference models

Figure 2: (a) **Consistent generative models on finite sets**. Observed variables are shown in white, and hidden variables are shaded. Stochastic paths are indicated with dashed lines while deterministic paths are solid lines. Conditioned on the function inputs $x_{1:n}$ and the auxiliary latent variables $z^{(1:T)}$, the function outputs are transformed independently using instance-wise conditional normalising flows (CNFs). If the initial states $\{(x_i, y_i)\}_{i=1:n}$ come from a SP, the construction will ensure that the collection of marginals $p(y_{1:n}^{(t)}|x_{1:n})$ are consistent under marginalisation for any $t$. (b) **Permutation-invariant inference models**. Auxiliary latent variables $z^{(1:T)}$ are inferred in reverse order in our inference model. We reuse the parameters of CNFs in the generative model to compute function values $y_{1:n}^t$ at all time steps, where each step has a different colour. We parameterise the conditional distribution $q_{x_{1:n}}^{(t)}(z^{(t)}|z^{(t+1)}, y_{1:n}^{(t)}; \phi)$ with permutation-invariant neural networks.

In Equation (12), the intermediate function outputs $y_{1:n}^{(1:T-1)}$ are not observed. However, we can sample them autoregressively by iterating between sampling $z^{t+1}$ and calculating $y_i^{(t)} = (F_\theta^{(t+1)})^{-1}(y_i^{(t+1)}; x_i, z^{(t+1)})$ for each $i$ (see Figure 2b). Therefore, we share the parameters of the normalising flows $F_\theta^{(1:T)}$ between the generative and inference models, leading to better performance. This idea was originally proposed by Cornish et al. [11], except that our normalising flows are applied to a sequence of function outputs $y_{1:n}$ given the inputs $x_{1:n}$.

In practice, the inference model $q_{x_{1:n}}(z^{(1:T)} \mid y_{1:n}; \phi)$ provides an approximate prior/posterior. Ideally, if it gives the exact posterior, conditional consistency would hold perfectly. However, when the inference model is approximate, the degree of conditional consistency depends on the discrepancy between the inference model and the true posterior. Predictive SP models also have a stochastic mapping conditioned on the context, represented as $q_{x_{1:m}}(z^{(1:T)} | y_{1:m}; \phi)$. However, it is important not to confuse this with the inference model, as they do not serve to approximate the posterior.

## 5 Related work

**Bayesian nonparametric models** Bayesian nonparametric models such as GPs [24, 48] and Student-t processes [5, 10, 52] provide common classical approaches for modelling distributions over functions. Under these models, any conditional distribution of a target given a context can be directly evaluated, and both marginal/conditional consistency is guaranteed by construction (if all computation is exact). However, they can be too restrictive for some applications, e.g. any conditional or marginal distribution of a GP is also Gaussian. Further, the evaluation of conditional densities is typically computationally intensive (cubic w.r.t. the context due to matrix inversion). Deep GPs [13, 59] use GPs as building blocks to construct deep architectures, designed to be flexible enough to model a wide range of complex systems. However, only the hyperparameters for each GP layer are tunable, which means they can still be restrictive when modelling highly non-Gaussian data.

**Copula-based processes** In multivariate statistics, a copula function refers to a multivariate function that describes the dependence between random variables and links the marginal distributions of each variable to the joint distribution of all the variables [28, 41]. Similarly, a copula process [58] describes the dependency between infinitely many random variables independent of their marginal distributions. [32, 33, 38] exploited this independence to specify the more flexible Copula-Based Processes (CBPs) by combing the copula processes of existing SPs with flexible models of marginal distributions based on normalising flows [16, 43, 49]; the consistency of CBPs is guaranteed as long as the base SPs are consistent. However, CBPs are still restrictive in terms of modelling relationship between variables because the underlying copula processes still come from known SPs. For example, BRUNOs [32, 33] have the same underlying copula processes as GPs, so they cannot be used to model data with non-Gaussian dependencies.

**Neural process family** NPs [22] are generative SP models which specify a prior SP and rely on Bayesian inference to compute conditional densities. To improve expressivity of the original NPs, [4, 18, 30] explore predictive SP models that directly learn mappings from context to predictive SPs. More specifically, ANPs [30] incorporate cross-attention modules to model the interaction between context and target. CONVNPs [18] produce a functional representation with a convolutional architecture which parameterizes a stationary predictive SP over the target, given the context. In GNPs [4], predictive SPs are modelled by GPs where the mean/kernel functions are directly produced by neural networks conditioned on the context. However, simply setting the context to an empty set in these models does not yield more expressive generative SP models. In the absence of context, ANPs and GNPs become standard NPs and GPs respectively, without any additional expressivity. CONVNPs can indeed be adjusted for generative SP models where the latent variables are *random functions*. However, it remains unclear how to perform Bayesian inference in function space [18]. Conditional NP families, e.g. Conditional Neural Processes (CNPs) [21], Convolutional Conditional Neural Processes (CONVCNPs) [25] are another category of predictive SP models, which make a strong assumption that all targets are independent given the context. Recently, Transformer Neural Processes (TNPs) [42], Neural Diffusion Processes (NDPs) [17] provide an alternative to the NP models above, showing promising predictive performance. However, both marginal and conditional consistency are sacrificed in these models.

Table 1: **Estimated marginal log-likelihood of SPs on 1D function regression problems**. We compare (a) Oracle models (when available). (b) GPs with optimised hyperparameters for additive kernels with three component kernels including an RBF kernel, a Matern kernel and a periodic kernel. (c) CBPs which combine Gaussian copula processes with neural spline flows [16]. (d) NPs. (e) MNPs with 7 transition steps. Each experiment is repeated 5 times and we report the mean and standard errors of the marginal log-likelihood normalised by the number of points in the target. When latent models such as MNPs, NPs are used, we obtain estimations of marginal log-likelihoods on test data using the IWAE objective [6] with 20 latent samples.

| Model | Samples from GPs | | | Non-GP Data | | |
|---|---|---|---|---|---|---|
| | RBF Kernel | Matern Kernel | Periodic Kernel | Monotonic | Convex | SDEs |
| Oracle | $2.846_{\pm 0.012}$ | $2.709_{\pm 0.013}$ | $0.641_{\pm 0.006}$ | — | — | — |
| GPs | $\mathbf{2.844}_{\pm 0.013}$ | $\mathbf{2.708}_{\pm 0.014}$ | $0.419_{\pm 0.013}$ | $0.633_{\pm 0.059}$ | $2.976_{\pm 0.224}$ | $1.719_{\pm 0.034}$ |
| CBPs | $2.628_{\pm 0.016}$ | $2.604_{\pm 0.015}$ | $0.169_{\pm 0.022}$ | $1.776_{\pm 0.088}$ | $4.268_{\pm 0.035}$ | $1.842_{\pm 0.024}$ |
| NPs | $0.935_{\pm 0.019}$ | $1.115_{\pm 0.021}$ | $0.356_{\pm 0.020}$ | $1.823_{\pm 0.006}$ | $1.956_{\pm 0.004}$ | $1.621_{\pm 0.009}$ |
| MNPs | $2.491_{\pm 0.024}$ | $2.290_{\pm 0.021}$ | $\mathbf{0.594}_{\pm 0.032}$ | $\mathbf{2.755}_{\pm 0.010}$ | $\mathbf{5.582}_{\pm 0.081}$ | $\mathbf{1.942}_{\pm 0.019}$ |

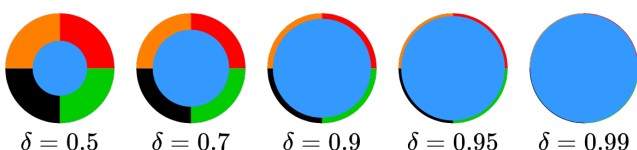

$\delta = 0.5 \qquad \delta = 0.7 \qquad \delta = 0.9 \qquad \delta = 0.95 \qquad \delta = 0.99$

Figure 3: Wheel contextual bandits with varying exploration parameter $\delta$. The optimal actions are $a_1, \ldots, a_5$ when the context point are in blue, yellow, red, green and black regions respectively.

## 6 Experiments

Our experiments aim to answer the questions: 1) Can the proposed framework of MNPs offer better expressivity than NPs? 2) Compared to Bayesian nonparametric models, do neural parameterised SPs have advantages, especially on highly non-Gaussian data? 3) How well do MNPs perform on scientific problems? All experiments are performed using PyTorch [44]. For details about datasets, please see Appendix B.1. For additional experimental details such as hyperparameters and architectures, please refer to Appendix B.2 and our reference implementation at https://github.com/jinxu06/mnp.

### 6.1 1D function regression

We first consider 1D function regression problems to perform controlled comparisons between GPs, CBPs, NPs, and MNPs. Table 1 shows results across a range of datasets, including samples from the GP prior (we consider three different kernels) as well as more challenging non-GP data.

For datasets sampled from GPs, we also include the performance of oracle GPs with the right hyperparameters for generating the data. As shown, GPs and CBPs are close to the oracle except for samples generated using a periodic kernel, where CBPs struggle to learn the right kernel hyperparameters due to the difficulty of optimisation. For neural parameterised models, the performance gaps between the MNPs and the oracle GPs are much smaller than for NPs.

For non-GP samples such as monotonic functions, convex functions and samples from certain SDEs, MNPs perform the best across all models. CBPs obtain better marginal log-likelihood than GPs because the pointwise marginals are transformed using normalising flows, but their underlying copula processes (which capture the dependence between data points) are still Gaussian, restricting their ability to model highly non-Gaussian data compared to neural parameterised models e.g. MNPs, NPs.

### 6.2 Contextual bandits

Contextual bandits are a class of problems where agents repeatedly choose from a set of actions based on context and receive rewards. The goal is then to learn a policy that maximizes expected cumulative reward over time. Contextual bandits find applications in real-time decision making problems such as resource allocation and online advertising.

Table 2: **Results on wheel contextual bandit problems**. We use an increasing value of $\delta$, where more exploration is needed with higher $\delta$. We report the mean and standard deviation of both cumulative and simple regrets (a performance measure of the final policy, estimated by the mean cumulative regrets in the last 500 steps) over 100 trials. Results are normalised by the performance of a uniform agent which chooses actions with equal probability.

| $\delta$ | 0.5 | 0.7 | 0.9 | 0.95 | 0.99 |
|---|---|---|---|---|---|
| *Cumulative regrets* | | | | | |
| Uniform | $100.0_{\pm 0.00}$ | $100.0_{\pm 0.00}$ | $100.0_{\pm 0.00}$ | $100.0_{\pm 0.00}$ | $100.0_{\pm 0.00}$ |
| MAML | $2.95_{\pm 0.12}$ | $3.11_{\pm 0.16}$ | $4.84_{\pm 0.22}$ | $7.01_{\pm 0.33}$ | $22.93_{\pm 1.57}$ |
| NPs | $1.60_{\pm 0.06}$ | $1.75_{\pm 0.05}$ | $3.31_{\pm 0.10}$ | $5.71_{\pm 0.24}$ | $22.13_{\pm 1.23}$ |
| MNPs | $\mathbf{1.08}_{\pm 0.00}$ | $\mathbf{1.23}_{\pm 0.01}$ | $\mathbf{2.10}_{\pm 0.01}$ | $\mathbf{2.07}_{\pm 0.01}$ | $\mathbf{5.46}_{\pm 0.05}$ |
| *Simple regrets* | | | | | |
| Uniform | $100.0_{\pm 0.00}$ | $100.0_{\pm 0.00}$ | $100.0_{\pm 0.00}$ | $100.0_{\pm 0.00}$ | $100.0_{\pm 0.00}$ |
| MAML | $2.49_{\pm 0.12}$ | $3.00_{\pm 0.35}$ | $4.75_{\pm 0.48}$ | $7.10_{\pm 0.77}$ | $22.89_{\pm 1.41}$ |
| NPs | $\mathbf{1.04}_{\pm 0.06}$ | $\mathbf{1.26}_{\pm 0.21}$ | $2.90_{\pm 0.35}$ | $5.45_{\pm 0.47}$ | $21.45_{\pm 1.3}$ |
| C-BRUNO | $1.32_{\pm 0.06}$ | $1.43_{\pm 0.07}$ | $3.44_{\pm 0.13}$ | $6.17_{\pm 0.21}$ | $21.52_{\pm 0.41}$ |
| MNPs | $\mathbf{1.14}_{\pm 0.06}$ | $\mathbf{1.29}_{\pm 0.08}$ | $\mathbf{2.12}_{\pm 0.16}$ | $\mathbf{2.24}_{\pm 0.16}$ | $\mathbf{5.22}_{\pm 0.38}$ |

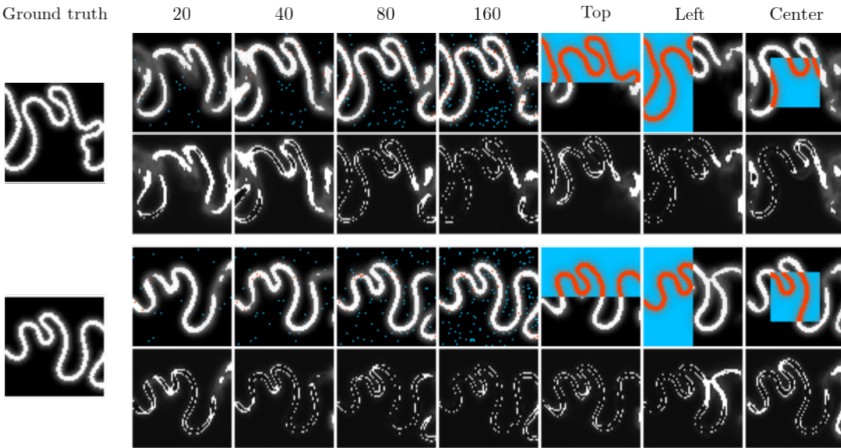

Figure 4: **Inferred geology conditioned on measurements**. Given a small number of physical measurements (red pixels indicate positive measurements while blue pixels indicate negative ones), we show the predictive mean (first row in each panel) and standard deviation (second row in each panel). Different columns correspond to different context sets. As can be seen, MNPs make predictions that are consistent with the data while showing larger uncertainty when fewer context points are available (or when a prediction is made far from a context point).

Similarly to Garnelo et al. [22], we use the wheel bandit problem [50] (see Appendix B.1) to evaluate our approach: a unit circle is partitioned into 5 regions (see Figure 3) whose sizes are controlled by an exploration parameter $\delta$. There are 5 actions $a_1, a_2, a_3, a_4, a_5$, and their associated reward depend on a 2D contextual coordinate $X = (X_1, X_2)$ uniformly sampled from within the circle. If $||X|| \leq \delta$, $a_1$ is the optimal action with reward sampled from $\mathcal{N}(1.2, 0.01^2)$, and taking any other action would yield a reward $r \sim \mathcal{N}(1.0, 0.01^2)$. If $||X|| > \delta$, the optimal action depends on which of the remaining 4 region $X$ falls into and choosing it would yield a reward $r \sim \mathcal{N}(50, 0.01^2)$. Under this circumstance, any other action yield a reward $\mathcal{N}(1.0, 0.01^2)$ except that $a_1$ receives $r \sim \mathcal{N}(1.2, 0.01^2)$. We follow the experimental setup from Garnelo et al. [22] and only include models with a pre-training procedure (see Appendix B.2 for details). As can be seen in Table 2, MNPs significantly outperform baselines (taken from the results of [22]) across different exploration rates $\delta$.

### 6.3 Geological inference

In geostatistics, one is often interested in inferring the geological structure of an area given only a sparse set of measurements. This problem has traditionally been tackled with variants of GP regression

Table 3: **Test marginal log-likelihood on geology data**. Uniform context points are uniformly sampled from within the 2D square, while regional context includes the top/left half or the central square with half the size.

| $\|\mathcal{C}\|$ | Uniform Context | | | | Regional Context | | |
|---|---|---|---|---|---|---|---|
| | $N = 20$ | 40 | 80 | 160 | Top | Left | Center |
| NPs | $-0.35_{\pm 0.30}$ | $-0.31_{\pm 0.29}$ | $-0.28_{\pm 0.27}$ | $-0.25_{\pm 0.27}$ | $-39.63_{\pm 304.08}$ | $-22.03_{\pm 107.25}$ | $-1.18_{\pm 4.20}$ |
| MNPs | $\mathbf{0.67}_{\pm 0.31}$ | $\mathbf{0.80}_{\pm 0.23}$ | $\mathbf{0.98}_{\pm 0.18}$ | $\mathbf{1.12}_{\pm 0.15}$ | $\mathbf{0.44}_{\pm 0.59}$ | $\mathbf{0.22}_{\pm 0.61}$ | $\mathbf{0.65}_{\pm 0.39}$ |

[12] (often referred to as Kriging in this context). However, several geological patterns (e.g. fluvial patterns) are highly complex and cannot be properly captured by these methods. To address this, several geostatistical models use a single training image to extract geological patterns and match these to the measurements [53, 63]. However, these approaches generally fail to produce realistic patterns capturing the variability of real geology.

More recently, deep learning approaches have been applied to this problem. [8, 14, 62] train GANs on large geological datasets and use this as a prior for inferring geological structure given sparse measurements. However, the resulting models do not provide any meaningful uncertainty estimates, which are crucial for decision making in several applications. As MNPs provide uncertainty estimates they may therefore be a compelling approach for this problem.

To evaluate MNPs on this problem, we introduce the GeoFluvial dataset, containing more than 25k simulations of fluvial patterns as $128 \times 128$ grayscale images. The dataset was generated using the open source `meanderpy` [54] package, which itself simulates the geological patterns as SPs (see Appendix B.1 for details). We train our model on a training set of 20k simulations and evaluate it on a test set of 5k simulations. We test our trained model on a variety of context point configurations, corresponding to geological measurements taken at various spatial locations. Specifically, we consider uniformly sampled measurements (at 20, 40, 80, and 160 locations) as well as scenarios where measurements are spatially restricted to a certain area (top, left, and in the center of the square). Quantitative results are shown in Table 3 and qualitative results in Figure 4. As can be seen, our model achieves better likelihoods than NPs on this problem for all context point configurations, while also generating patterns that match the geological context. Further, as can be seen in Figure 4, the uncertainty estimates of the model are consistent with expectations, i.e. the model is more uncertain far from measurement locations or when there is ambiguity in the direction of the fluvial pattern.

## 7   Discussion

**Limitations and future work**   Because we apply only a finite number of Markov transitions, the entire computational graph containing intermediate states is held in memory for backpropagation. To alleviate this, an interesting direction for future work would be to consider continuous-time Markov transitions, i.e. stochastic differential equations in function space, which require less memory by using adjoint methods [9, 37]. While Equation (9) provides a valid general construction of exchangeable and consistent MMTOs with latent variables, it may be feasible to investigate other forms of MMTOs that do not necessitate latent variables, thereby eliminating the need for variational approaches.

There are often inherent symmetries in data (e.g. translational symmetries for stationary SPs) and it may be possible to improve empirical performance of MNPs using (group equivariant) convolutions that preserve certain symmetries [18, 25, 29]. These improvements are, however, orthogonal to our contributions and combining them is an exciting direction for future work.

**Conclusions**   We have introduced Markov Neural Processes (MNPs), a framework for constructing flexible neural generative SP models. MNPs use neural-parameterized Markov transition operators on function spaces to gradually transform a simple initial SP into a flexible one. We proved that our proposed neural transitions preserve exchangeability and consistency necessary to define valid SPs. Empirical studies demonstrate that we can obtain expressive models of SPs with promising performance on contextual bandits and geological data.

## Acknowledgments and Disclosure of Funding

We would like to thank Hyunjik Kim for valuable discussions. We also thank Peter Tilke for useful discussions around generating the geological dataset. JX gratefully acknowledges funding from Tencent AI Labs through the Oxford-Tencent Collaboration on Large Scale Machine Learning.

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

# Appendix of Deep Stochastic Processes via Functional Markov Transition Operator

## A  Proofs

### A.1  Proof of proposition 4.1 (See page 4)

**Proposition 4.1.** *Let $F_\theta(\cdot; x, z) : \mathcal{Y} \mapsto \mathcal{Y}$ denote an invertible transformation between outputs, parameterized by the input and latent. If $\{p_{x_{1:n}}(y_{1:n}^{(0)})\}_{x_{1:n}}$ is a valid Stochastic Process (SP) (i.e. it satisfies Equation (1) and Equation (2)) and*

$$y_{1:n}^{(0)} \sim \{p_{x_{1:n}}(y_{1:n}^{(0)})\}_{x_{1:n}}, \quad z \sim p_\theta(z), \quad y_i = F_\theta(y_i^{(0)}; x_i, z). \tag{6}$$

*then $\{p_{x_{1:n}}(y_{1:n})\}_{x_{1:n}}$ is also a valid SP.*

*Proof.* Given an input $x_i$ and a latent variable $z$, $y_i$ is obtained by applying an invertible transformation $F_\theta$ to $y_i^{(0)}$. Therefore, we can represent $p(y_i|y_i^{(0)}; x_i, z)$ using the Dirac delta:

$$p(y_i|y_i^{(0)}; x_i, z) = \delta(y_i - F_\theta(y_i^{(0)}; x_i, z)) \tag{16}$$

Furthermore, according to Equation (6), these transformations are independently applied given the latent variable $z$. So we have:

$$p(y_{1:n}|y_{1:n}^{(0)}; x_{1:n}, z) = \prod_{i=1}^{n} \delta(y_i - F_\theta(y_i^{(0)}; x_i, z)) \tag{17}$$

Hence

$$p_{x_{1:n}}(y_{1:n}|y_{1:n}^{(0)}) = \int p_\theta(z) p_\theta(y_{1:n}|y_{1:n}^{(0)}; x_{1:n}, z) \mathrm{d}z \tag{18}$$

$$= \int p_\theta(z) \prod_{i=1}^{n} \delta(y_i - F_\theta(y_i^{(0)}; x_i, z)) \mathrm{d}z \tag{19}$$

$\square$

As per Proposition 4.3, $\{p_{x_{1:n}}(y_{1:n}|y_{1:n}^{(0)})\}_{x_{1:n}}$ is exchangeable and consistent. Furthermore, because $\{p_{x_{1:n}}(y_{1:n}^{(0)})\}_{x_{1:n}}$ is a valid SP, $\{p_{x_{1:n}}(y_{1:n})\}_{x_{1:n}}$ is also a valid SP

### A.2  Proof of proposition 4.2 (See page 4)

**Proposition 4.2.** *If the collection of marginals before transition $\{p_{x_{1:n}}(y_{1:n}^{(0)})\}_{x_{1:n}}$ is consistent and exchangeable (as per Equations (1) and (2)) and the collection of marginal Markov transition operators (MMTOs) $\{p_{x_{1:n}}(y_{1:n}|y_{1:n}^{(0)})\}_{x_{1:n}}$ is also consistent and exchangeable (as per Equations (7) and (8)), then the collection of marginals after transition $\{p_{x_{1:n}}(y_{1:n})\}$ is also consistent and exchangeable, hence defining a valid SP.*

*Proof.* According to the definitions of consistency and exchangeability for both the collection of marginals $\{p_{x_{1:n}}(y_{1:n}^{(0)})\}_{x_{1:n}}$ and the collection of transition operators $\{p_{x_{1:n}}(y_{1:n}|y_{1:n}^{(0)})\}_{x_{1:n}}$, we have the following (Equations (1), (2), (7) and (8)):

$$\int p_{x_{1:n}}(y_{1:n}^{(0)}) \mathrm{d}x_{m+1:n} = p_{x_{1:m}}(y_{1:m}^{(0)})$$

$$p_{\pi(x_{1:n})}(\pi(y_{1:n}^{(0)})) = p_{x_{1:n}}(y_{1:n}^{(0)})$$

$$\int p_{x_{1:n}}(y_{1:n}|y_{1:n}^{(0)}) \, \mathrm{d}y_{m+1:n} = p_{x_{1:m}}(y_{1:m}|y_{1:m}^{(0)})$$

$$p_{x_{1:n}}(y_{1:n}|y_{1:n}^{(0)}) = p_{\pi(x_{1:n})}(\pi(y_{1:n})|\pi(y_{1:n}^{(0)}))$$

where $1 < m < n$.

The Markov transition on the function outputs are described by:

$$p_{x_{1:n}}(y_{1:n}) = \int p_{x_{1:n}}(y_{1:n}^{(0)})p_{x_{1:n}}(y_{1:n}|y_{1:n}^{(0)})dy_{1:n}^{(0)}$$

For any finite sequence $x_{1:n}$, we have

$$\int p_{x_{1:n}}(y_{1:n})dy_{m+1:n} = \int \left( \int p_{x_{1:n}}(y_{1:n}^{(0)})p_{x_{1:n}}(y_{1:n}|y_{1:n}^{(0)})dy_{1:n}^{(0)} \right)dy_{m+1:n}$$

$$= \int p_{x_{1:n}}(y_{1:n}^{(0)})\left( \int p_{x_{1:n}}(y_{1:n}|y_{1:n}^{(0)})dy_{m+1:n} \right)dy_{1:n}^{(0)}$$

$$= \int p_{x_{1:n}}(y_{1:n}^{(0)})p_{x_{1:m}}(y_{1:m}|y_{1:m}^{(0)})dy_{1:n}^{(0)}$$

$$= \int \left( \int p_{x_{1:n}}(y_{1:n}^{(0)})dy_{m+1:n}^{(0)} \right)p_{x_{1:m}}(y_{1:m}|y_{1:m}^{(0)})dy_{1:m}^{(0)}$$

$$= \int p_{x_{1:m}}(y_{1:m}^{(0)})p_{x_{1:m}}(y_{1:m}|y_{1:m}^{(0)})dy_{1:m}^{(0)}$$

$$= \int p_{x_{1:m}}(y_{1:m})dy_{1:m}^{(0)} \tag{20}$$

Furthermore, we have

$$p_{\pi(x_{1:n})}(\pi(y_{1:n})) = \int p_{\pi(x_{1:n})}(\pi(y_{1:n}^{(0)}))p_{\pi(x_{1:n})}(\pi(y_{1:n})|\pi(y_{1:n}^{(0)}))d\pi(y_{1:n}^{(0)})$$

$$= \int p_{x_{1:n}}(y_{1:n}^{(0)})p_{x_{1:n}}(y_{1:n}|y_{1:n}^{(0)})dy_{1:n}^{(0)}$$

$$= p_{x_{1:n}}(y_{1:n}) \tag{21}$$

Therefore, the collection of marginals $\{p_{x_{1:n}}(y_{1:n})\}_{x_{1:n}}$ are both consistent and exchangeable (Equations (20) and (21)), hence defining a valid SP.

$\square$

### A.3 Proof of proposition 4.3 (See page 4)

**Proposition 4.3.** MMTOs *in the form of Equation* (9) *are consistent and exchangeable.*

*Proof.* Recall that MMTOs in Equation (9) write as:

$$p_{x_{1:n}}(y_{1:n}|y_{1:n}^{(0)};\theta) = \int p_\theta(z)\prod_{i=1}^{n}\delta(y_i - F_\theta(y_i^{(0)};x_i,z))\,dz, \tag{22}$$

It is a special case of a more general form:

$$p_{x_{1:n}}(y_{1:n}|y_{1:n}^{(0)};\theta) = \int p_\theta(z)\prod_{i=1}^{n}p_\theta(y_i\,|\,y_i^{(0)},x_i,z)dz, \tag{23}$$

Equation (23) becomes Equation (22) when $p_\theta(y_i\,|\,y_i^{(0)},x_i,z) = \delta(y_i - F_\theta(y_i^{(0)};x_i,z))$ is a $\delta$-distribution. Below we prove that MMTOs are consistent and exchangeable for the general form. We have

$$\int p_{x_{1:n}}(y_{1:n}|y_{1:n}^{(0)};\theta)dy_{m+1:n} = \int \left( \int p_\theta(z)\prod_{i=1}^{n}p_\theta(y_i\,|\,y_i^{(0)},x_i,z)\,dz \right)dy_{m+1:n}$$

$$= \int p_\theta(z)\prod_{i=1}^{m}p_\theta(y_i\,|\,y_i^{(0)},x_i,z)\left( \int \prod_{i=m+1}^{n}p_\theta(y_i\,|\,y_i^{(0)},x_i,z)dy_{m+1:n} \right)dz$$

$$= \int p_\theta(z)\prod_{i=1}^{m}p_\theta(y_i\,|\,y_i^{(0)},x_i,z)\,dz$$

$$= p_{x_{1:m}}(y_{1:m}|y_{1:m}^{(0)};\theta) \tag{24}$$

and

$$p_{\pi(x_{1:n})}(\pi(y_{1:n})|\pi(y_{1:n}^{(0)}); \theta) = \int p_\theta(z) \prod_{i=\pi(1)}^{\pi(n)} p_\theta(y_i \mid y_i^{(0)}, x_i, z)\mathrm{d}z$$

$$= \int p_\theta(z) \prod_{i=1}^{n} p_\theta(y_i \mid y_i^{(0)}, x_i, z)\mathrm{d}z$$

$$= p_{x_{1:n}}(y_{1:n}|y_{1:n}^{(0)}; \theta) \qquad (25)$$

Therefore, the collection $\{p_{\pi(x_{1:n})}(\pi(y_{1:n})|\pi(y_{1:n}^{(0)}); \theta)\}_{x_{1:n}}$ is both consistent (Equation (24)) and exchangeable (Equation (25)). $\qquad\square$

## A.4   Derivation of Markov Neural Process marginal densities

$T$ step Markov Neural Processes (MNPs) have marginal densities in the following form (as in Equation (10)):

$$p_{x_{1:n}}(y_{1:n}; \theta) = \int p_\theta(z^{(1:T)})p_{x_{1:n}}(y_{1:n}^{(0)}) \prod_{t=1}^{T}\prod_{i=1}^{n} p_\theta^{(t)}(y_i^{(t)} \mid y_i^{(t-1)}, x_i, z^{(t)})\mathrm{d}y_{1:n}^{(0:T-1)}\mathrm{d}z^{(1:T)}$$

$$= \int p_\theta(z^{(1:T)})p_{x_{1:n}}(y_{1:n}^{(0)}) \prod_{t=1}^{T}\prod_{i=1}^{n} \left|\det\frac{\partial F_\theta^{(t)}(y_i^{(t-1)}; x_i, z^{(t)})}{\partial y_i^{(t-1)}}\right|\mathrm{d}z^{(1:T)}$$

*Proof.* Given the sequence of latent variables $z^{(1:n)}$, our model becomes normalising flows on the finite sequence of function outputs $y_{1:n}$, with a prior distribution $p_{x_{1:n}}(y_{1:n}^{(0)})$, and the invertible transformation at each step $F_\theta^{(t)}(\cdot; x_i, z^{(t)})$. According to Papamakarios et al. [43], Rezende and Mohamed [49], we have

$$p_{x_{1:n}}(y_{1:n}|z^{(1:T)}; \theta) = p_{x_{1:n}}(y_{1:n}^{(T)}|z^{(1:T)}; \theta)$$

$$= p_{x_{1:n}}(y_{1:n}^{(T-1)}|z^{(1:T-1)}; \theta) \prod_{i=1}^{n} \left|\det\frac{\partial F_\theta^{(T)}(y_i^{(T-1)}; x_i, z^{(T)})}{\partial y_i^{(T-1)}}\right|$$

$$= \cdots$$

$$= p_{x_{1:n}}(y_{1:n}^{(0)}) \prod_{t=1}^{T}\prod_{i=1}^{n} \left|\det\frac{\partial F_\theta^{(t)}(y_i^{(t-1)}; x_i, z^{(t)})}{\partial y_i^{(t-1)}}\right|$$

The marginal density $p_{x_{1:n}}(y_{1:n}; \theta)$ can be computed as:

$$p_{x_{1:n}}(y_{1:n}; \theta) = \int p_\theta(z^{(1:T)})p_{x_{1:n}}(y_{1:n}|z^{(1:T)}; \theta)\mathrm{d}z^{(1:T)}$$

$$= \int p_\theta(z^{(1:T)}; \theta)p_{x_{1:n}}(y_{1:n}^{(0)}) \prod_{t=1}^{T}\prod_{i=1}^{n} \left|\det\frac{\partial F_\theta^{(t)}(y_i^{(t-1)}; x_i, z^{(t)})}{\partial y_i^{(t-1)}}\right|\mathrm{d}z^{(1:T)} \qquad (26)$$

$\qquad\square$

# B   Implementation details

## B.1   Data

### B.1.1   1D synthetic data

**Gaussian Process (GP) samples**   Three 1D function datasets were created, each comprising samples from Gaussian processes (GPs) with different kernel functions: RBF (length scale 0.25), Matern-2.5 (length scale 0.5), and Exp-Sine-Squared (length scale 0.5 and periodicity 0.5). Observation noise

variances were set at $0.0001$ for RBF and Matern kernels, and $0.001$ for the Exp-Sine-Squared kernel. Function inputs spanned from $-2.0$ to $2.0$. To accelerate sampling, identical input locations were employed for every 20 function instances. For this dataset, context size varies randomly from 2 to 50.

**Monotonic functions** Our generation of monotonic functions starts by sampling $N \sim \mathrm{Poisson}(5.0)$ to determine the number of interpolation nodes. We then sample $N+1$ increments $X_{\mathrm{increments}}$ sampled from a Dirichlet distribution. These increments are increased by $0.01$ to avoid excessively small values, and are then normalized such that their sum is $4.0$. The final $X$ values for interpolation nodes are obtained by adding $-2.0$ to the cumulative sum of these increments so that these $X$ values are within the range $[-2.0, 2.0]$. For each $X$ value, a corresponding $Y$ value is sampled from a Gamma distribution $Y \sim \mathrm{Gamma}(2, 1)$. The cumulative sum of $Y$ values ensures monotonicity. A PCHIP interpolator [20] is then created using these interpolation nodes ($X$ and $Y$ values) to generate function outputs. Given the functions, we randomly sample $128$ $X$ values and compute their corresponding function values. Note that these $X$ values are now used to evaluate the functions, rather than serving as interpolation nodes. The function values are normalized to the range $[-1.0, 1.0]$. Finally, Gaussian observation noise with a standard deviation of $0.01$ is added to these function values. For this dataset, context size varies randomly from 2 to 20.

**Convex functions** To create a dataset of convex functions, we compute integrals of the monotonic functions previously created. These convex functions are then randomly shifted and rescaled to increase diversity. The function values are normalized to the range $[-1.0, 1.0]$. Finally, Gaussian observation noise with a standard deviation of $0.01$ is added to these function values. Context sizes varied randomly from 2 to 20.

**Stochastic differential equations samples** We create a dataset of 1D functions, each of which represents a solution to a Stochastic Differential Equation (SDE). This SDE is defined by the drift function $f(x, t) = -(a + x \cdot b^2) \cdot (1 - x^2)$ and the diffusion function $g(x, t) = b \cdot (1 - x^2)$, with constants $a$ and $b$ both set to $0.1$. The function sets up a time span that includes $128$ uniformly distributed points within the range of $[-5.0, 5.0]$. We then uniformly sample an initial condition, $x_0$, between $0.2$ and $0.6$. We use the `sdeint.stratKP2iS` function from the `sdeint` library to generate a solution to the SDE. This solution forms a 1D function that depicts a trajectory of the SDE across the defined time span, originating from the initial condition $x_0$. Lastly, we randomly alter the context sizes between 2 and 50.

For all the aforementioned datasets, we use the following set sizes: 50000 for the training set, 5000 for the validation set, and 5000 for the test set.

### B.1.2  Geological data

We generate the GeoFluvial dataset using the `meanderpy` [54] package. We first run simulations using the numerical model of meandering in `meanderpy` with default parameters except for channel depth which we change from 16m to 6m. The resulting simulations correspond to images of shape (800, 4000). We then extract 3 random non-overlapping crops of shape (700, 700) from these images, which are resized to (128, 128) and are used as data. We ran $\lceil 25000/3 \rceil$ simulations, resulting in a dataset of 25,000 images.

### B.2  Model architectures and hyperparameters

**Permutation equivariant/invariant neural networks on sets** We implement two versions of neural modules which operate on sets, both of which preserve the permutation symmetry of the sets. They are known as deep sets [61] and set transformers [36]. To obtain an invariant representation, we used sum pooling for deep sets, and pooling by multi-head attention (PMA) [36] for set transformers. Our experiments primarily employed set transformers, chosen for their stronger ability to model interactions between datapoints. However, for the wheel contextual bandit experiments, the context set often expanded to tens of thousands. To circumvent memory issues, we use deep sets in this instance. For set transformers, we stack two layers of set attention blocks (SABs) with a hidden dimension of $64$ and $4$ heads. This is followed by a single layer of PMA, which was subsequently followed by a linear map. In the case of using deep sets, we use a shared instance-wise Multi-Layer Perceptron (MLP) that has two hidden layers and a hidden dimension of $64$. This MLP processes the concatenation of function inputs and outputs. Following this, we add a sum aggregation which is then succeeded by a single-layer MLP with ReLU (Rectified Linear Unit) activation.

**Conditional normalising flows** The instance-wise invertible transformation $F_\theta^{(t)}$ at each time step $t$ is parameterised as a rational quadratic spline flow [16]. Note that we do not share parameters among iterations. To condition on an input $x_i$ and a latent variable $z$, we use a Multi-Layer Perceptron (MLP) which takes in $x_i$, $z$ and produces the parameters for configuring a one-layer spline flow with 10 bins.

**Multi-Layer Perceptrons (MLPs)** Except for the deep sets, we use MLPs to parameterise continuous functions in two places. Firstly, we use it as a conditioning component in conditional normalising flows $F_\theta^{(t)}$ (as we mentioned above). It has two hidden layers and a hidden dimension of 128. Secondly, we use it to parameterise $\mu_\phi^{(t)}$ and $\sigma_\phi^{(t)}$ in the inference model (see Section 4.3). It has two hidden layers and a hidden dimension of 64.

We adopt the same pretraining approach as Garnelo et al. [22] for the contextual bandits problem, pretraining the model on 64 wheel problems $\{\delta_i\}_{i=1}^{64}$, where $\delta_i \sim \mathcal{U}(0,1)$. Each wheel $\delta_i$ has a context size ranging from 10 to 512 and a target size varying between 1 and 50. Here each data point is a tuple $(X, a, r)$. Because the context size can grow to tens of thousands during test time, for computational efficiency, we use deep sets to implement SETENCODER and a linear flow rather than a spline flow to parameterise $F_\theta^{(t)}$.

For optimisation, we use Adam [31] optimiser with a learning rate of 0.0001. We use a batch size of 100 for 1D synthetic data and a batch size of 20 for the geological data. In experiments, we find that it is often beneficial to encode $x_i$ with Fourier features [55], and we use 80 frequencies randomly sampled from a standard normal.

For more details, please also see our reference implementation at `https://github.com/jinxu06/mnp`.

### B.3 Computational costs and resources

In our current implementation, the invertible transformations $F_\theta^{(t)}$ in the generative model and the mean/variance function $\mu_\phi^{(t)} \sigma_\phi^{(t)}$ in the inference model do not share parameters across iterations. However, we do share the SETENCODER across iterations. Consequently, with our MNP, both memory usage and computing time increase linearly with the number of steps. If the SETENCODER employs set transformers, the computational cost becomes $O(m^2)$, where $m$ stands for the context size. This cost, however, can be decreased to $O(mk)$ by replacing set attention blocks (SABs) with induced set attention blocks (ISABs), where $k$ denotes the number of inducing points. If deep sets are used to implement SETENCODER, the computational cost reduces to $O(m)$, despite the inference model being less expressive. The computatiional advantage of using deep sets becomes prominent when we have a large number of observed datapoints in the context. For instance, in the contextual bandits experiments (in Section 6.2), our model needs to process 80K datapoints, which is intractable for a transformer-based NP on a single GPU.

To further enhance scalability, one might consider sharing some parameters across transition steps. Additionally, in the case of space complexity, as discussed in Section 7, we can explore continuous-time Markov transitions, i.e., stochastic differential equations in function space. Such an approach would potentially require less memory by leveraging adjoint methods.

Training MNP is indeed resource-intensive; however, inference in MNP simply requires a forward pass. On a single GeForce GTX 1080 GPU card, a standard 7-step MNP takes approximately one day to train for $200k$ steps on 1D functions. If we use 20 latent samples to evaluate the marginal log-likelihood using the IWAE objective [6], inference runs typically in a few seconds for a batch of 100 functions.

## C Ablation studies

The better performance of MNPs compared to standard Neural Processes (NPs) can roughly be attributed to two reasons: Firstly, instead of directly constructing the SP as in NPs, MNPs iteratively transform from a trivial SP gradually to a more expressive one. Secondly, MNPs employ more expressive invertible transformations on function outputs, parameterized by normalizing flows. To verify that multiple steps are indeed advantageous, we evaluated various models based on their estimated marginal log-likelihood on GP data using an RBF kernel. The results are as follows:

- MNP, 1 step, spline flow: 0.941
- MNP, 2 steps, spline flow: 1.950
- MNP, 3 steps, spline flow: 2.266
- MNP, 1 step, spline flow, latent dimension $\times 2$: 0.850
- MNP, 1 step, spline flow, latent dimension $\times 2$, MLP hidden dimension $\times 2$: 1.476
- MNP, 1 step, linear transformation: 0.990
- MNP, 2 steps, linear transformation: 1.977
- MNP, 3 steps, linear transformation: 2.140

From these results, we observe that performance generally improves with an increase in the number of steps. Using higher-dimensional latent variables or wider neural networks in models with only one step doesn't achieve the performance seen in models with multiple steps. Furthermore, replacing the spline flow with a linear transformation (keeping in mind that stacking multiple linear transformations remains linear) still showed a consistent performance improvement with an increase in steps.

## D    Broader impacts

Our work presents MNPs, a novel approach to construct SP using neural parameterised Markov transition operators. The broader impacts of this work have many aspects. Firstly, the proposed models are more flexible and expressive than traditional SP models and NPs, enabling them to handle more complex patterns that arise in many applications. Moreover, the exchangeability and consistency in MNPs could improve the robustness and reliability of neural SP models. This could lead to more trustworthy systems, which is a critical aspect in high-stakes applications. However, as with any machine learning system, there are potential risks. For example, the complexity of these models could exacerbate issues related to interpretability and transparency, making it more difficult for humans to understand and control their behaviour.

