# OpenReview forum: "Deep Stochastic Processes via Functional Markov Transition Operators"
_NeurIPS.cc/2023/Conference — NeurIPS 2023 poster_

### Official Review · Reviewer_UK93 · 2023-06-20

**Soundness:** 2 fair
**Presentation:** 1 poor
**Contribution:** 3 good
**Rating:** 6
**Confidence:** 3

**Summary:**

The paper introduces a new class of Bayesian nonparametric  models by extending neural processes. The fundamental idea is to create a Markov chain of stochastic processes, culminating in a flexible enough stochastic process.

**Strengths:**

The paper introduces a simple but a clever modification of vanilla neural processes. The idea allows for more expressive representation as demonstrated by the experiments. I believe the idea is significant enough for it to be of interest to the community.

**Weaknesses:**

The paper uses an unclear notation for probabilistic notions. It’s not just that this choice is unconventional, forcing the reader to keep a mental map of how things are redefined, but also that it needlessly complicates exposition. For example, what is $p$? It is ubiquitous in the paper but is never clearly defined. This makes the discussion too hand-wavy — what is $p_{x_{1:n}}(y_{1:n})$? Is it the distribution function of the random vector $(F(x_1), \ldots, F(x_n))$? If so, what even is $y_{1:n}$? Is the density function of the random vector $(F(x_1), \ldots, F(x_n))$? What is $f$? Since $\mathcal X$ is simply the indexing set, and for real-valued random variables we should have $\mathcal Y = \mathbb R$, is $f$ simply a real-valued function defined on the indexing set? The notation for transition operators is also very troublesome. My point in raising such questions is that I would have preferred if the authors used standard measure-theoretic notation (it is standard for a good reason) to write much of the paper (I really like the treatment in Ghosal and van der Vaart’s book).

**Questions:**

None

**Limitations:**

Not applicable.

---

> ### Author Rebuttal · Authors · 2023-08-09
>
> We are grateful for the reviewer’s constructive feedback and have provided our responses below:
>
> > *“The paper uses an unclear notation for probabilistic notions. It’s not just that this choice is unconventional, forcing the reader to keep a mental map of how things are redefined, but also that it needlessly complicates exposition.”*
>
> > *“I would have preferred if the authors used standard measure-theoretic notation (it is standard for a good reason) to write much of the paper (I really like the treatment in Ghosal and van der Vaart’s book).”*
>
> We appreciate the reviewer's suggestions in terms of notation. While the referenced book indeed offers a nice notation system based on measure theory, we are worried about several potential issues that could arise if we alter our notation:
> * Our current notation aims to align as closely as possible with the original neural process (NP) paper [1]. This system is widely adopted within the NP community, and adopting a different one may cause confusion to some readers.
> * We intentionally write everything using density functions so that it is more accessible to the more application-oriented reader. For example, we denote stochastic processes (SPs) as a collection of density functions $\\{p_{x_{1:n}}(y_{1:n})\\}\_{x_{1:n}}$. In this context, '$p$' represents densities over a finite sequence $y_{1:n}$ in the traditional sense. It is indexed by input sequences $x_{1:n}$ because each input sequences correspond to a different density. Furthermore, we express the marginal transition operators $p(y_{x_{1:n}}^{(t)}|y_{x_{1:n}}^{(t-1)})$ as a conditional density function, avoiding the use of operator notations. However, we appreciate that a measure-theoretic notation might be preferable to readers with a more theoretical background.
>
> Below, we provide a more detailed explanation of our notation, which we hope will improve clarity. We are entirely open to further suggestions on improving our notation while maintaining consistency with [1] and accommodating our non-theoretical audience.
>
> [1] Garnelo, Marta, et al. "Neural processes." arXiv preprint arXiv:1807.01622 (2018).
>
> > *“What is p_{x_{1:n}}(y_{1:n})? Is it the distribution function of the random vector (F(x_1), F(x_2), …, F(x_n))? If so, what even is y_{1:n}? Is the density function of the random vector (F(x_1), F(x_2), …, F(x_n))? ? What is f?”*
>
> Every $p$ in our paper is a density function in the traditional sense, representing a probability distribution. We use the colon notation to denote a sequence, i.e. $x_{1:n}=[x_1,x_2,\dots,x_n]$, $y_{1:n}=[y_1,y_2,\dots,y_n]$ (see Equation 1 in our paper and Equation 1 in the NP paper [1]). We will add a few sentences to our paper to make it clear.
>
> $F$ represents a random function, while $f$ stands for a specific realization of that function. This is analogous to the convention in Euclidean space where a random variable is often symbolized by a capital letter, such as X. The density function, denoted as $p(x)$, describes the probability of X assuming the value of $x$, where $x$ is a specific realization of the variable. In a similar vein, $[F(x_1), F(x_2), \dots, F(x_n)]$ denotes a sequence of random variables, resulting from evaluating the random function $F$ at respective points $x_i$. The sequence $y_{1:n}=[f(x_1), f(x_2), \dots, f(x_n)]$ represents the specific realization of this sequence.
>
> > *“Since $\mathcal{X}$ is simply the indexing set, and for real-valued random variables we should have $\mathcal{Y}=\mathbb{R}$, is $f$ simply a real-valued function defined on the indexing set?”*
>
> In our paper, we treat functions as infinite-dimensional vectors (each element corresponds to a function value at a certain point). A stochastic process then becomes a distribution in the infinite-dimensional space. Through this lens, the outputs of functions are considered random variables, while their inputs serve as the continuous indices for these variables. Essentially, in our context, the term 'indexing set' is synonymous with the function input domain.
>
> > *“The notation for transition operators is also very troublesome.”*
>
> Transition operators represent a conditional distribution of the next states given the current states. Therefore, we express all transition operators in terms of their corresponding conditional distributions, without introducing the operator notation.
>
> * Marginal Transition Operators: $p_{x_{1:n}}(y_{1:n}^{(t)} | y_{1:n}^{(t-1)})$ denotes the conditional density function of the function outputs at time $t$ given the function outputs at time $t-1$. The subscript $x_{1:n}$ indicates that these conditional density functions are different for different input sequences $x_{1:n}$.
>
> * Functional Transition Operators: In our paper, we use $p(f^{(t)})$ to denote a stochastic process (i.e. a distribution over functions). Thus, the functional transition operator $p(f^{(t)} | f^{(t-1)})$ refers to the conditional distribution of the current functions (a stochastic process) given the prior one. In our paper, functional transition operators are defined indirectly via marginal transition operators. The notation here is only for ease of discussion.

---

> > ### Comment · Reviewer_UK93 · 2023-08-18
> >
> > Thank you very much for the clarifications. It makes a lot more sense and I have updated my rating to reflect that. In particular, I do not want to force my views on use of measure-theoretic notation if the rest of the literature is not following that at all.

---

### Official Review · Reviewer_wA6N · 2023-07-07

**Soundness:** 3 good
**Presentation:** 2 fair
**Contribution:** 3 good
**Rating:** 6
**Confidence:** 3

**Summary:**

This paper proposes Markov neural process, a new class of stochastic processes. The work is based on prior art on neural process, which was published in 2018. The paper is reasonably written. The effectiveness of the new method is very clear from the experimentations.

**Strengths:**

1. The proposed markov neural process adds more expressiveness to the traditional neural process, which potentially can open the door to new applications;
2. The theoretical analysis appears sounds
2. The proposed method appears to be competitive when compared to the baselines.

**Weaknesses:**

1. The explanation of the overall MNP is hard to follow. Fig 2 is crucial to explain the overall structure, plus the training and inference flow. But it's unclear. For example, the three type of the arrows in Fig 2(b) aren't explained well.
2. The experiments are impressive in the tables. But the manuscript didn't provide any intuitive explanation why MNP works better. In the contextual bandit example, MNP shows significant improvement over NP when $\delta = 0.99$. Some explanation the reason or an ablation study with respect to the number of markov stages is needed for the readers to get a better feeling.

**Questions:**

1. please elaborate the statement "the exchangeability and consistency in MNPs could improve the robustness and reliability of neural SP model".

**Limitations:**

No potential negative societal impact.

---

> ### Author Rebuttal · Authors · 2023-08-09
>
> We thank the reviewer for dedicating their time to evaluating our manuscript. Below are our responses:
>
> > *“The explanation of the overall MNP is hard to follow. Fig 2 is crucial to explain the overall structure, plus the training and inference flow. But it's unclear. For example, the three type of the arrows in Fig 2(b) aren't explained well.”*
>
> We have revised Figure 2 and included the updated version in the uploaded 1-page PDF above. We believe that the figure is now much clearer.
>
> > *“The experiments are impressive in the tables. But the manuscript didn't provide any intuitive explanation why MNP works better.”*
>
> The better performance of MNPs can be attributed to the following reasons:
> * Instead of directly constructing the SP as in NPs, MNPs iteratively transform from a trivial SP gradually to a more expressive one.
> * MNPs employ more expressive invertible transformations on function outputs, parameterized by normalizing flows.
>
> These two aspects correspond to the extensions mentioned in Section 4.1, transitioning from NPs to a more generalized form, which serves as the foundation for MNPs. We will include these discussions in the experimental section.
>
> > *“Some explanation the reason or an ablation study with respect to the number of markov stages is needed for the readers to get a better feeling”*
>
> To address this point, we ran a simple ablation study. We compared MNPs with different numbers of Markov transitions in terms of the estimated marginal log-likelihood on GP data with a RBF kernel:
> * MNP (T=1), spline flows, 0.941
> * MNP (T=2), spline flows, 1.950
> * MNP (T=3), spline flows, 2.266
>
> As the results show, in this setting the performance improves as the number of steps increases. We will run a more extensive ablation study and include the detailed results in the next version of the paper.
>
> > *“Please elaborate the statement "the exchangeability and consistency in MNPs could improve the robustness and reliability of neural SP model"*
>
> This statement appears in the broad impact section in the appendix. While there are many recent developments to construct neural SP models, we highlight that many of these models do not consider conditional consistency, which is problematic, especially for sequential decision-making. There is also the potential for such models to fall foul of the so-called Dutch Book scenarios. Additionally, models such as NDPs do not meet marginal consistency, so the distribution over functions is not properly defined, and conflicting predictions can be made.
>
> MNPs, on the other hand, provide a framework to extend the original NPs and enhance their expressivity without compromising consistency. When using MNPs for predictions, there's a guarantee that the predictions won’t contradict each other and will hence be more reliable. MNPs also enjoy better generalization because of the consistency properties, and are less sensitive to changes to the data such as the observation order in sequential decision-making. In this regard, MNPs contribute to improving the robustness and reliability of neural SP models.

---

> > ### Comment · Reviewer_wA6N · 2023-08-15
> > **I've read the rebuttal**
> >
> > Thanks for the updated figure. It's much clearer than the previous version.
> >
> > > As the results show, in this setting the performance improves as the number of steps increases. We will run a more extensive ablation study and include the detailed results in the next version of the paper.
> >
> > I would strongly encourage the authors to complete a more comprehensive ablation study in the revision.

---

### Official Review · Reviewer_kRke · 2023-07-07

**Soundness:** 3 good
**Presentation:** 3 good
**Contribution:** 3 good
**Rating:** 7
**Confidence:** 3

**Summary:**

The paper attempts to improve the flexibility of Neural Processes without sacrificing exchangeability and consistency properties of classical stochastic processes. The paper proposes a diffusion-like iterative refinement of the predicted values, however, ensuring exchangeability and consistency.

**Strengths:**

- Theoretically ensures exchangeability and consistency.
- Performs better than the main baseline in this category i.e., NPs.

**Weaknesses:**

I discuss this in the Questions section.

**Questions:**

1. I feel Eq. (11) is somehow incomplete considering that $y_i^{(t)}$ (for $t=0, \ldots, T-1$) are also random variables that do not appear here. Are they absorbed in the reconstruction term (i.e., the first term)?
2. Is there an empirical way to test how big of a problem marginal consistency poses, for instance, by measuring KL between LHS and RHS of Eq. (2). What I am getting at is: Is marginal consistency really a big issue empirically speaking, especially considering the flexibility of the state of the art models like TNPs or NDPs.
3. The paper does not cite or discuss Transformer Neural Processes (TNPs). Perhaps one may also compare with Neural Diffusion Processes (NDPs). It may help position the performance of MNPs even if it cannot outperform these considering that these models relax the consistency requirement.
4. One aspect I think is that MNPs do not require pairwise interaction between the query points required by the transformer-based NP architectures. I wonder if one might expect some computational benefits of MNPs with respect to this class of models.
5. It would be nice to provide a derivation of the second line of equation (10).

**Limitations:**

Yes, the paper discusses the limitations.

---

> ### Author Rebuttal · Authors · 2023-08-09
>
> We are grateful to the reviewer for their detailed comments. Below are our responses:
>
> > *“I feel Eq. (11) is somehow incomplete considering that y_i^{(t)} (for t=0,..., T-1) are also random variables that do not appear here. Are they absorbed in the reconstruction term (i.e., the first term.”*
>
> Thanks for pointing out the confusion here. Note that our model is intentionally constructed such that, while $y_{1:n}^{(t)}$ are indeed random variables, they become deterministic functions of $y_{1:n}^{(t-1)}$ once the corresponding latent variables $z^{(t)}$ are provided (see Eq. (9)). This construction ensures that while the Markov transitions from $y_{1:n}^{(t-1)}$ to $y_{1:n}^{(t)}$ are stochastic, all the stochasticity is encapsulated in the latent variables $z^{(t)}$. Consequently, there is no need to integrate over $y_{1:n}^{(t-1)}$ in Eq. 11. This design choice is crucial in making our model practical by requiring integration only over low-dimensional latent variables with variational inference.
>
> In Eq. (11), when the sequence of auxiliary latent variables $z^{(1:T)}$ is provided, the sequence of transition steps from $y_{1:n}^{(0)}$ to $y_{1:n}^{(T)}$ become deterministic. Therefore, $p(y_{m+1:n} | z^{(1:T)})$ can be directly computed with a change of variables, in the same way as normalizing flows. The primary difference is that now the invertible transformations are applied to a set rather than a single vector. Hence, the intermediate outputs transitioning from $y_{1:n}^{(t-1)}$ to $y_{1:n}^{(t)}$ are indeed absorbed in the first term, as they become deterministic variables once all the latent variables $z^{(1:T)}$ are provided.
>
> To provide a better illustration of the generative process, we also updated Figure 2 (Please see the uploaded PDF above).
>
> > *“Is marginal consistency really a big issue empirically speaking, especially considering the flexibility of the state-of-the-art models like TNPs or NDPs.”*
>
> There are several practical reasons why marginal consistency may be desirable - we highlight some of these below.
>
> * Data Efficiency: Models lacking consistency guarantees, such as TNPs and NDPs, require direct learning of consistent properties from the data, potentially making these models more data-hungry than consistent SP models.
> * Out-of-Distribution Generalization: Without consistency guarantees, model properties become entirely dependent on the training data distribution. As an example, if the model consistently encounters target sizes greater than $M$ during training, its performance is likely to deteriorate when faced with tasks where the target size is less than $M$. Conversely, models maintaining marginal consistency are expected to naturally generalize to smaller target sizes.
> * Predictive Reliability: When using MNPs for predictions, there's an assurance that the predictions won’t contradict each other. However, such reliability cannot be guaranteed with TNPs and NDPs.
>
> We are planning to verify the aforementioned conjectures via experiments, but we may not be able to finish them within the rebuttal period.
>
> Generally speaking, the choice of model largely depends on the application. If the sole objective is to attain the best possible fit to the data, then models without consistency or exchangeability properties, such as TNPs and NDPs, may be more appropriate. However, if the goal is to construct a model that offers better generalizability, interpretability, and robustness, models that have both exchangeability and (marginal and conditional) consistency properties might be preferred.
>
> > *“The paper does not cite or discuss Transformer Neural Processes (TNPs). Perhaps one may also compare with Neural Diffusion Processes (NDPs).”*
>
> Thanks for providing this reference. We will cite and discuss TNPs in our related work section. We will also discuss NDPs in more detail in the related work section.
>
> > *“One aspect I think is that MNPs do not require pairwise interaction between the query points required by the transformer-based NP architectures. I wonder if one might expect some computational benefits of MNPs with respect to this class of models.”*
>
> The reviewer's observation is entirely accurate. When we use deep sets to parameterize our permutation-invariant set encoders in the inference model (line 176), our model scales linearly with respect to the context size. In contrast, for transformer-based NP architectures, the cost is quadratic in the context size. This advantage becomes prominent when we have a large number of observed datapoints in the context. For instance, in the contextual bandits experiments (Sec 6.2), our model needs to process 80K datapoints, which is intractable for a transformer-based NP on a single GPU.
>
> > *“It would be nice to provide a derivation of the second line of equation (10).”*
>
> The detailed derivation for equation 10 can be found in Appendix A.4.
>
> In particular, from the first to the second line of Equation 10, it's crucial to recognize that while both $y_{1:n}^{0:T-1}$ and $z^{(1:T)}$ are random variables, once the latent variables $z^{(1:T)}$ and the initial states of the function outputs $y_{1:n}^{0}$ are conditioned, $y_{1:n}^{1:T-1}$ becomes deterministic (see Equation 9). For deterministic transformations, the resulting density functions can be computed as the product of the original density functions and the determinant of the inverse Jacobian of the transformations, and we do not need to integrate over intermediate variables.

---

> > ### Comment · Reviewer_kRke · 2023-08-17
> > **Thank You for the Response**
> >
> > I read the rebuttal and I thank the authors for their response.
> >
> > Thanks for the updated figure and clarification about the deterministic dependencies. It helps a lot.
> >
> > I appreciate the clarification that certain applications may require consistency as a guarantee rather than as a may-be. It makes sense.
> >
> > I have revised my rating.

---

### Official Review · Reviewer_nP7k · 2023-07-11

**Soundness:** 3 good
**Presentation:** 3 good
**Contribution:** 3 good
**Rating:** 8
**Confidence:** 3

**Summary:**

The paper presents Markov Neural Processes (MNPs), a novel class of Stochastic Processes (SPs) that leverage neural networks for data modeling. MNPs enhance the flexibility and expressivity of the Neural Processes (NPs) framework without sacrificing consistency or imposing restrictions. The proposed iterative construction is fully generative and maintains consistency under conditioning. MNPs demonstrate superior performance over baseline models across various tasks.

The primary contribution is the MNP model, which employs a neural process as the transition operator, thereby adding flexibility to the stochastic process. This method increases expressivity while preserving the fully generative nature and consistency of NPs. The conditional consistency defined in this paper plays a crucial role in preventing uncertainty mismatches.

**Strengths:**

he model is theoretically proven to satisfy the valid conditions. The experiments are well-designed to demonstrate the model's better expressivity, advantage over traditional models on non-Gaussian data, and better performance on scientific problems. The method has high theoretical novelty, especially in the use of Markov operators that lifts the Gaussian restrictions of the previous models.

**Weaknesses:**

The experiments are run on a few small-scale synthetic datasets. The model's performance would be better evaluated if there are experiments on a larger variety of, especially large real-world, datasets. Besides, experiments that evaluates the time consumption among the models would be helpful.



**Questions:**

How is the scalability of this method? Were any tricks used to make the method more scalable? Can you provide an analysis on the time/space complexity?

**Limitations:**

Yes.

---

> ### Author Rebuttal · Authors · 2023-08-09
>
> We deeply appreciate the time and effort the reviewer has dedicated to evaluating our paper. Please see below for our responses to the questions raised.
>
> > *“How is the scalability of this method? Were any tricks used to make the method more scalable? Can you provide an analysis on the time/space complexity?”*
>
> If we employ deep sets to parameterize the permutation-invariant set encoders in the inference model (line 176), MNPs with $T$ steps are roughly $T$ times more expensive than the original neural processes in terms of both time and space complexity. However, it's important to note that MNPs still represent a scalable and expressive model compared to attention-based models such as ANPs and transformer neural processes: Our model scales linearly with respect to the context size. In contrast, for transformer-based NP architectures, the cost is quadratic in the context size. This advantage becomes prominent when we have a large number of observed datapoints in the context. For instance, in the contextual bandits experiments (Sec 6.2), our model needs to process 80K datapoints, which is intractable for a transformer-based NP on a single GPU.
>
> To further enhance scalability, one might consider sharing some parameters across transition steps. Additionally, in the case of space complexity, as discussed in Section 7, we can explore continuous-time Markov transitions, i.e., stochastic differential equations in function space. Such an approach would potentially require less memory by leveraging adjoint methods.
>
> When making predictions with trained $T$-step MNPs, the time complexity with deep sets parameterized set encoders stands at $\mathcal{O}(BT(N_{\mathcal{C}}+N_{\mathcal{T}}))$ where $B$ is batch size, $N_{\mathcal{C}}$ is context size (the observed number of datapoints) and $N_{\mathcal{T}}$ is target size (the predicted number of datapoints). In contrast, with set transformers parameterized set encoders, it is $\mathcal{O}(BT(N_{\mathcal{C}}+N_{\mathcal{T}})+BN_{\mathcal{C}}^2)$. Note that the set transformer is used to compute a representation shared across steps so there is no multiplier of $T$ for the quadratic term. The space complexity follows a similar pattern: $\mathcal{O}(BT(N_{\mathcal{C}}+N_{\mathcal{T}}))$ for deep sets and $\mathcal{O}(BT(N_{\mathcal{C}}+N_{\mathcal{T}})+BN_{\mathcal{C}}^2)$ for set transformers. It's worth noting that in this analysis, we've excluded the complexity related to the depth and width of each MLP component for simplicity.

---

### Official Review · Reviewer_ipFc · 2023-07-26

**Soundness:** 4 excellent
**Presentation:** 4 excellent
**Contribution:** 4 excellent
**Rating:** 8
**Confidence:** 4

**Summary:**

The paper introduces Markov Neural Processes (MNPs), a new variant of Stochastic Processes (SP). The paper points out that predictive SP models that construct context-to-target mappings no longer satisfy conditional consistency property and also are not fully generative. An expressive NP variant called MNP is proposed, the basis of which is stacking sequences of Markov transition operators in the function space. The paper adapts the definitions of consistency and exchangeability to MNPs, and show that they are consistent and exchangeable by construction. The paper shows the potential of MNPs in a variety of benchmarks from 1D function regression to geological inference.


**Strengths:**

The paper introduces the framework from a very generic point of view, and also provides context in the specific choices made for the experiments. For instance, the usage of instance-wise conditional normalizing flows to parameterize the MNP is very interesting; as flows are universal approximators, this gives the MNPs high flexibility. Similarly, the usage of permutation-invariant neural networks is also interesting.

The idea is simple but powerful, and the exposition of the idea is very neat.


**Weaknesses:**

The empirical evaluation of the paper is somewhat limited. More empirical results may attract readers to this paper - for instance, few-shot image classification results would be interesting to see. Results in more large-scale datasets will also be useful.


**Questions:**

Figure 2: I could not understand clearly why the auxiliary latent variables should be inferred in the reverse order. Can you expand further?

The paper uses a specific variant of normalizing flows in its experiments called spline flows. It would be useful to justify the usage of the specific variant of flows. More flexible variants such as deep sigmoidal flows can also be tried out.


**Limitations:**

The training of MNPs is computationally intensive as the authors state; this limitation can be addressed in future works.

---

> ### Author Rebuttal · Authors · 2023-08-09
>
> We sincerely thank the reviewer for their feedback on our manuscript. Below are our responses to the questions raised.
>
> > *“Figure 2: I could not understand clearly why the auxiliary latent variables should be inferred in the reverse order. Can you expand further?”*
>
> The auxiliary latent variables are inferred in reverse order primarily to enable parameter sharing between the generative model and the inference model. During inference, function outputs $y_{1:n}^{(T)}$ are observed. Upon inferring $z^{(T)}$ first, we can directly compute $y_{1:n}^{(T-1)}$ by inverting the transformation $F_{\theta}^{(T)}$ which is already learned in the generative model. Recursively, for other steps, $F_{\theta}^{(t)}$ can be reused in the inference model in a similar way. This design was initially proposed by [1] and is a critical aspect that contributes to the effectiveness of their methods.
>
> Moreover, it should be noted that the prior over auxiliary latents $z^{(1:T)}$ can be specified in an arbitrary order. Therefore, we can express $p(z^{(1:T)})$ as $p(z^{(1:T)})=p(z^{(T)})p(z^{(T-1)} | z^{(T)}) \cdots p(z^{(1)} | z^{(2:T)})$. This concept is illustrated in Figure 2(a) (which is updated and uploaded in the 1-page PDF above); specifically in the direction of the arrows for the latent variables. As a result, the inferred posterior over the latent variables can actually have the same factorization as the prior over these latent variables.
>
> [1] Cornish, Rob, et al. "Relaxing bijectivity constraints with continuously indexed normalising flows." International conference on machine learning. PMLR, 2020.
>
> > *“The paper uses a specific variant of normalizing flows in its experiments called spline flows. It would be useful to justify the usage of the specific variant of flows. More flexible variants such as deep sigmoidal flows can also be tried out.”*
>
>
> Applying different normalizing flows to our model would indeed be very interesting. We did not exhaustively optimize our model and it may be possible that other normalizing flows perform better. Our main motivation for using spline flows is that they are very expressive and flexible even on low-dimensional spaces (including in 1D) - as our normalizing flows operate on the space of $\mathcal{Y}$ which is often low-dimensional, this made spline flows a simple yet effective choice for our purposes. Compared to spline flows, many advanced normalizing flows excel particularly in high-dimensional domains. That said, more suited and powerful alternatives to spline flows may indeed exist and it would be interesting to explore this further in future work.

---

> > ### Comment · Reviewer_ipFc · 2023-08-16
> >
> > Thank you for the clarification.
> >
> > My score remains the same.

---

### Author Rebuttal · Authors · 2023-08-09

Here we updated Figure 2 in the uploaded PDF, which illustrates the generative and inference models of MNPs. In the reviewer rebuttals below, we address each reviewer's questions and comments separately.

---

### Decision · Program_Chairs · 2023-09-21

**Decision:**

Accept (poster)

**Comment:**

This paper proposes a new class of stochastic processes called Markov Neural processes that stacks sequences of Markov transition operators in the function space and satisfies the requirements of proper stochastic processes. There is agreement among the reviewers on the novelty and technical strength of the paper as well as in its limited experimental evaluation. Overall, I believe the first two aspects (novelty and technical strength) are contributions worth of presentation at NeurIPS.